# Phytochemical Profile of Antibacterial Agents from Red Betel Leaf (*Piper crocatum* Ruiz and Pav) against Bacteria in Dental Caries

**DOI:** 10.3390/molecules27092861

**Published:** 2022-04-30

**Authors:** Leny Heliawati, Seftiana Lestari, Uswatun Hasanah, Dwipa Ajiati, Dikdik Kurnia

**Affiliations:** 1Study Program of Chemistry, Faculty of Mathematics and Natural Science, Universitas Pakuan, Bogor 16143, Indonesia; seftiana.062118062@unpak.ac.id (S.L.); uswatun.hasanah@unpak.ac.id (U.H.); 2Department of Chemistry, Faculty of Mathematics and Natural Science, Universitas Padjadjaran, Sumedang 45363, Indonesia; dwipa20001@mail.unpad.ac.id (D.A.); dikdik.kurnia@unpad.ac.id (D.K.)

**Keywords:** red betel leaf, *Piper crocatum* Ruiz and Pav, antibacterial, *Streptococcus mutans*, phytochemical profiling

## Abstract

Based on data from The Global Burden of Disease Study in 2016, dental and oral health problems, especially dental caries, are a disease experienced by almost half of the world’s population (3.58 billion people). One of the main causes of dental caries is the pathogenesis of *Streptococcus mutans*. Prevention can be achieved by controlling *S. mutans* using an antibacterial agent. The most commonly used antibacterial for the treatment of dental caries is chlorhexidine. However, long-term use of chlorhexidine has been reported to cause resistance and some side effects. Therefore, the discovery of a natural antibacterial agent is an urgent need. A natural antibacterial agent that can be used are herbal medicines derived from medicinal plants. *Piper crocatum* Ruiz and Pav has the potential to be used as a natural antibacterial agent for treating dental and oral health problems. Several studies reported that the leaves of *P. crocatum* Ruiz and Pav contain secondary metabolites such as essential oils, flavonoids, alkaloids, terpenoids, tannins, and phenolic compounds that are active against *S. mutans*. This review summarizes some information about *P. crocatum* Ruiz and Pav, various isolation methods, bioactivity, *S. mutans* bacteria that cause dental caries, biofilm formation mechanism, antibacterial properties, and the antibacterial mechanism of secondary metabolites in *P. crocatum* Ruiz and Pav.

## 1. Introduction

The oral cavity is a place of growth for more than 700 species of microorganisms, which ultimately has many impacts on the health of the teeth and oral cavity. One of the health problems experienced globally is oral infectious diseases such as dental caries [1,2,3]. In 2017, the prevalence of dental caries in permanent teeth per 100,000 population in each country reached 20% to more than 50% [4]. The cause is the synergistic interaction of bacteria such as *Streptococcus sanguinis* and *S. mutans* to form a biofilm on the tooth surface [5,6,7,8,9]. The high prevalence of dental caries and the weakness of the strategies used today indicate an urgent need to identify alternative treatment options that are more effective and efficient, one of which is the use of medicinal plants [10].

Some studies reported that red betel leaf has the potential to be used as a natural antibacterial agent in treating dental and oral health problems. Red betel leaf contains secondary metabolites such as essential oils, flavonoids, alkaloids, and phenolic compounds that actively inhibit *S. mutans* [11,12]. Based on this, this review focuses on the antibacterial activity found in red betel leaf (*P. crocatum* Ruiz and Pav) which has been studied extensively [13]. This review will also discuss the relationship between antibacterial activity and the structure of several compounds contained in red betel leaf extract.

## 2. Gram-Positive and Negative Bacteria Cause Dental Caries

### 2.1. Gram-Positive Bacteria

#### 2.1.1. *Streptococcus mutans*

*S. mutans* is a Gram-positive bacterium that is considered to be the microorganism that most often plays a role in tooth decay [14]. These bacteria are able to organize themselves in the bacterial community through cell–cell interactions and connections with other components present in the medium such as polysaccharides, proteins, and DNA to form biofilms [15,16]. Biofilm is a structured and organized community of microbial cells in a dynamic environment, enclosed and embedded in a three-dimensional (3D) extracellular matrix [17,18,19]. The cariogenic biofilm matrix formed by *S. mutans* is rich in exopolysaccharides and contains extracellular DNA (eDNA) and lipoteichoic acid (LTA) [20,21,22,23]. Microbial species are found in oral biofilms such as *Candida albicans*, *Candida glabrata*, *Enterococcus faecalis*, *S. mutans*, *Veillonella dispar*, *Fusobacterium nucleatum,* and many others [24].

One of the diseases caused by *S. mutans* is dental caries. There are several factors that cause dental caries to get worse including sugar, saliva, and also putrefactive bacteria [25,26,27]. In addition, the growth of bacteria in the mouth and forming biofilms is caused by several factors, namely saliva which plays a role in modulating the plaque layer on the teeth, the temperature in the environment around the mouth in the range of 35–36 °C, and pH 6.75–7.25 [28,29]. The mechanism of biofilm formation on teeth is followed by five stages, namely initial adhesion which produces extracellular polymeric substances, initial attachment where cell division occurs, formation of young biofilms, mature biofilms, and dispersed biofilms which cause cell autolysis [30] (Figure 1).

The pathogenesis of *S. mutans* begins after consuming something containing sugar, especially sucrose, a sticky glycoprotein (a combination of protein and carbohydrate molecules) that is retained on the teeth to initiate plaque formation on the teeth [31,32]. At the same time, millions of bacteria, including *S. mutans*, also survive on the glycoprotein. *S. mutans* has an enzyme called glucosyl transferase on its surface which is involved in glycolysis [25,33,34]. Glycolysis is the breaking down of glucose in sucrose that is carried out to obtain energy.

The glucosyltransferase enzyme continues to work, namely, to add more glucose molecules to form dextran which has a structure very similar to amylase in starch. Dextran together with other bacteria adheres tightly to the tooth enamel and subsequently forms plaque on the teeth [35,36]. In addition, glycolysis under anaerobic conditions also produces lactic acid. This lactic acid causes a decrease in pH to a certain extent so that it can destroy hydroxyapatite in the tooth enamel and cause the formation of a cavity or hole in the tooth [37,38] (Figure 2).

#### 2.1.2. *Streptococcus sanguinis*

*Streptococcus sanguinis* is a type of Gram-positive bacteria that does not have spores and is a facultative anaerobe. Cell division in *S. sanguinis* occurs along a single axis and produces chains or pairs of cocci. The genome sequence of *S. sanguinis* SK36 isolated from dental plaque in humans has a circular DNA molecule consisting of 2,388,435 base pairs, with 2274 predicted protein codes. In tRNA, there are 61 genes that are predicted to be able to produce 20 amino acids and 50 carbohydrate transporters, including the phosphotransferase enzyme which functions to transport glucose, fructose, mannose, cellobiose, glucoside, lactose, trehalose, galactitiol, and maltose. *S. sanguinis* is able to utilize various carbohydrate sources to survive [39].

Oral biofilm formation begins with the attachment of *S. sanguinis* and other pioneering colonists to a macromolecular complex formed on the saliva-coated tooth surface [22,40,41,42]. *S. sanguinis* was the first bacterium to bind to the biofilm and a species that plays an important role in the oral biofilm ecosystem [43,44,45,46]. However, these bacteria also have a positive role, namely producing H_2_O_2_ as a means to produce excess oxygen and working as a non-specific antimicrobial agent that can trigger the growth of *S. mutans* and other anaerobic periodontal pathogens [47,48,49]. 

The negatively charged residue and electrostatic interactions with hydrophilic regions in salivary proteins facilitate the attachment of bacteria to the tooth surface to form the Acquired Enamel Pellicle (AEP). Although *S. sanguinis* can directly adhere to saliva-free hydroxyapatite, the major mineral found in tooth enamel, the initial attachment process is most likely driven by the interaction of the streptococcal surface with salivary components. Binding to salivary proteins is mediated through protein–protein or protein–carbohydrate interactions with receptors exposed on the bacterial surface. Amylase is the most abundant salivary protein and is present both in AEP and in dental plaque. *S. sanguinis* specifically binds to amylase via long filamentous pili [50,51].

### 2.2. Gram-Negative Bacteria

####  *Veillonella*
*parvula*

*Veillonella parvula* is an anaerobic Gram-negative coccus that is part of the normal flora found in the human mouth and digestive tract [52]. Human oral *Veillonella* species include *V. parvula*, *V. dispar*, *V. atypica*, *V. denticariosi*, *V. rogosae*, *V. tobetsuensis*, *V. infantium,* and *V. nakazawae* [53,54,55]. Lactate and malate are the preferred carbon sources by *Veillonellae* spp. These carbon sources will be metabolized into propionate, acetate, CO_2_, and H_2_ [56,57]. Pyruvate, fumarate, and oxaloacetate can also be metabolized, but citrate, iso-citrate, and malonate are not. Succinate catabolism has been reported to have not resulted in suboptimal growth [58]. The balanced stoichiometry of lactate catabolism is (Equation (1)) [59]:8 Lactate → 5 Propionate + 3 Acetate + 3 CO_2_ + H_2_
(1)

Evidence that *Veillonellae* spp. acts as a linking species in biofilm development has been demonstrated in both in vivo and in vitro studies. Human epidemiological studies have shown *Veillonellae* spp. to be very abundant in both supra and sub-gingival plaques as well as on the tongue and in saliva [60,61,62,63,64]. *Veillonella* spp. (especially *V. parvula*) was found to be associated with dental caries in children [58,65]. Besides that, it was also found in adults. *V. parvula* was also one of the most abundant and prevalent bacteria in all samples of both healthy and carious teeth. However the abundance of *V. parvula* in carious tooth samples appears to be higher [66]. The physiological relationship between *Veillonellae* (as lactate users) and *S. mutans* (as lactate producers) has prompted many clinical studies on the relationship of *Veillonellae* with caries. Research conducted by Aas et al. [67] also demonstrated the association of the genera *Veillonella* with caries development. Belstrom et al. reported that *Streptococcus* spp. and *Veillonella* spp. were the most dominant genera among all saliva samples from 292 participants with mild to moderate dental caries [68].

It can be argued that the observed association between cariogenic bacteria and *Veillonella* stems from the metabolic need to produce organic acids which are indeed found in higher concentrations in active caries. Therefore, the presence of *Veillonellae* can be an indication of, and prediction of, a local decrease in pH. Bradshaw and Marsh reported that the number and proportion of *S. mutans* and *Lactobacillus* spp. increases as the pH decreases, especially below low pH [65]. Similarly in another clinical study, Gross et al. found the proportion of *Veillonellae* spp. increased commensurate with the proportion of *Streptococcus* spp. [69]. In other words, *Veillonellae* can be a risk factor for caries initiation, whereas *S. mutans* are a risk factor for caries development.

## 3. Antibacterial

### 3.1. Definition

An antibacterial is a substance that can inhibit the growth of bacteria and will kill pathogenic bacteria [70]. Antibacterial substances are divided into two types, namely bacteriostatic which suppresses bacterial growth and bactericidal which can kill bacteria [71]. Bacteria have evolved a lot to be able to survive in various environments and can develop resistance to various antibacterial reagents quickly [72]. Inhibition of bacteria can be through several synthesis pathways in bacteria, namely the bacterial cell wall biogenesis pathway, DNA replication pathway, transcription pathway, and protein biosynthesis pathway [73]. The cell wall structure consists of peptidoglycan which provides a mechanical effect on bacteria to maintain morphology. The peptidoglycan layer is formed from *N*-acetyl glucosamine and *N*-acetylmuramic acid linked by 1,4-glycosidic bonds [74].

### 3.2. Antibacterial Mechanism of Secondary Metabolic Compounds

Several secondary metabolites that are isolated from plants can be natural antibacterial agents. Each compound has their own antibacterial mechanism in inhibiting bacteria. Their mechanism will be explained in the following:

#### 3.2.1. Phenol

The mechanism of phenol as an antibacterial agent acts as a toxin in the protoplasm, damaging and penetrating the wall, causing the function of selective permeability, active transport, and protein composition control, so that bacterial cells become deformed and lysed [75,76,77].

#### 3.2.2. Flavonoids

Flavonoids work to inhibit bacterial growth by inhibiting nucleic acid synthesis, changing cytoplasmic membrane function, inhibiting energy metabolism, reducing cell attachment and biofilm formation, inhibiting porin in cell membranes, and disrupting permeability of cell walls and membranes to cause bacterial cell lysis [38,78,79,80,81]. In addition, flavonoids also act as inhibitors of the FabZ enzyme and inhibit the production of fimbriae [82].

#### 3.2.3. Saponins

Meanwhile, the saponins themselves work as antibacterial agents by disrupting the stability of the bacterial cell membrane, causing bacterial cell lysis [75,83,84,85].

#### 3.2.4. Terpenoids

Terpenoids work as antibacterials by disrupting the function of cell membranes to cause damage to bacterial cell membranes, interfering with glucosyltransferase activity, inactivating thiol-containing enzymes and causing bacterial death [86,87,88,89,90,91,92,93,94,95,96,97].

#### 3.2.5. Alkaloids

Alkaloids inhibit growth and kill bacteria by interfering with the permeability of cell walls and membranes, inhibiting of nucleic acid and protein synthesis, and inhibiting bacterial cell metabolism to cause lysis. Moreover, alkaloids can also act as inhibitors in the protein biosynthesis process in bacterial cells [98,99,100].

#### 3.2.6. Tannins

Tannins work by coagulating bacterial protoplasm, precipitating proteins, and binding proteins to inhibit the formation of bacterial cell walls [101,102,103] (Figure 3).

### 3.3. Antibacterial Mechanism with MurA Enzyme

In addition, the antibacterial mechanism can be carried out by inhibiting the action of the MurA enzyme that catalyzes the first step of bacterial cell wall biosynthesis. Therefore, the inhibition of the activity of oral pathogenic bacteria can be undertaken by inhibiting the enzyme MurA [104]. In cell wall peptidoglycan biosynthesis, the enzyme MurA involves the transfer of the enolpyruvate group from phosphoenolpyruvate (PEP) to UDP-N-acetylglucosamine (UNAG) to form UDP-N-acetylglucosamine enolpyruvate (UNAGEP) [90,91]. 

Based on the performance of fosfomycin, the inhibition of the MurA enzyme is competitive. Antibiotics act as PEP analogues and form covalent bonds with the active cysteine residue of the enzyme as shown in the figure below. Antibiotics interact with enzymes and UDP-N-acetylglucosamine and then form hydrogen bonds with different segments of the polypeptide chain. In addition, hydrogen bonds can be formed between the hydroxyl group of phosphomycin and the C-3 hydroxyl of the sugar ring UDP-N-acetylglucosamine and between one of its phosphonate oxygen atoms and the nitrogen amide of UDP-*N*-acetylglucosamine [105] (Figure 4).

### 3.4. Commonly Used Dental Caries Antibiotics

To control caries mediated by pathogenic bacteria, dental and oral hygiene products are widely used which consist of chemical compounds, such as fluoride, chlorhexidine, triclosan, cetylpyridinium chloride, and chlorophyll.

#### 3.4.1. Fluoride

Fluoride is the most effective caries prevention agent. Since the 1940s, it has been added to water supplies and oral care products, such as toothpaste, mouthwash, and dental floss [107]. In fact, the use of oral hygiene products containing fluoride reduced the prevalence of caries by 24–26% in permanent teeth. Water fluoridation in the range of 0.50–1.00 mg/L^−1^ is a cost-effective method for moderating caries potential [108]. In addition, the combination of nicomethanol hydrofluoride with siliglycol further enhances fluoride uptake by teeth and controls or inhibits dental biofilm development and strengthens tooth structure [109]. However, the use of fluoride for oral health also causes side effects, such as the emergence of fluoride-resistant strains [110,111]

#### 3.4.2. AIK(SO_4_)_2_

AIK(SO_4_)_2_ was found to be able to reduce fissure caries, both smooth surface and sulcus caries. The mechanism of dental caries treatment of alum may be almost the same as the mechanism of dental caries treatment using fluoride [112].

#### 3.4.3. Chlorhexidine (CHX)

Dental and oral hygiene products consist of another chemical compound, namely chlorhexidine (CHX). Chlorhexidine is a symmetric bis-biguanide agent consisting of two chloroguanide chains linked by a central hexamethylene chain and has diverse medical applications as a surface disinfectant and as an antiseptic for topical application. Chlorhexidine carryes two positive charges at physiological pH which can interact electrostatically with negatively charged phospholipids (CHX) and has been used to control dental caries caused by acid-tolerant bacteria such as *S. mutans* since the 1970s [113]. However, the use of chlorhexidine also causes certain disadvantages with long-term use such as tooth staining and taste changes [114]. It is also believed that the continued and increasing use of chlorhexidine can lead to the emergence of new strains of mycobacteria with lower susceptibility 

High prevalence of dental caries and the weakness of the strategies used today indicate an urgent need to identify alternative treatment options that are more effective, efficient, and non-toxic, one of which is by utilizing herbal medicines derived from medicinal plants [115]. In recent decades, research focus has also shifted to herbal medicines due to increasing bacterial resistance and side effects of antimicrobial agents. Extracts of plant origin can enhance antibiotic efficacy when used in combination against bacterial pathogens [10]. In addition, the use of medicinal plants or natural products is indeed a safe approach for rapid clinical translation because they are generally recognized as safe by the United States Food and Drug Administration.

## 4. *Piper crocatum* Ruiz and Pav

Based on some research literature, it has been reported that red betel leaf has the potential to be used as a natural antibacterial agent in treating dental and oral health problems. Red betel leaf (*P. crocatum* Ruiz and Pav) is a plant that grows in the tropics and was previously known as an ornamental plant, but was later used as a medicinal plant [116]. *P. crocatum* Ruiz and Pav is a natural ingredient that has the potential to treat dental caries and the leaf contains secondary metabolites such as essential oils, flavonoids, alkaloids, and phenolic compounds which may be active against *S. mutans* that plays a role in caries formation. The use of red *P. crocatum* Ruiz and Pav is traditionally useful in curing diseases such as canker sores and toothache. The red betel leaf decoction which is an antiseptic can act as a mouthwash, preventing bad breath. From chromatography it is known that *P. crocatum* Ruiz and Pav leaf contains flavonoid compounds, polyphenol compounds, tannins, and essential oils, where flavonoids are known to be inhibitors of the growth of *S. mutans* [11,50].

### 4.1. Isolation of Secondary Metabolites of Piper crocatum Ruiz and Pav

Several studies reported the isolation of *P. crocatum* Ruiz and Pav by many methods. Li et al., 2019 isolated 2.60 kg of dried red betel leaf samples, then extracted by reflux method using methanolic solvent (5 L × 3 times). The results of the isolation of *P. crocatum* Ruiz and Pav leaves revealed 23 compounds including 15 phenolic compounds (**1–15**), two monoterpenes (**16** and **17**), three sesquiterpene compounds (**19**–**21**), phenolic amide glycosides (**22**), neolignans (**23**), and the flavonoid compound C-glycoside (**24**). The structure of the compounds obtained was identified through spectroscopic methods and compared with the literature. Seven compounds (**7**, **11**, **13**, **14**, **17**, **20**, and **24**) of the species *P. crocatum* Ruiz and Pav and 17 others (**1**–**6**, **8**–**10**, **12**, **15**–**16**, **18**–**19**, and **21**–**23**) from the genus *Piper* and the family *Piperaceae* were isolated and reported for the first time [117] (Figure 5).

Another isolation method was carried out by Emrizal et al., 2014 for *P. crocatum* Ruiz and Pav, as much as 0.84 kg were extracted at room temperature with methanolic solvent to obtain a crude methanolic extract of 253.27 g (30.11%) after which the extract was evaporated, and they proceeded to separate the components of the compound. The results of the isolation obtained two compounds from the *P. crocatum* Ruiz and Pav plant which were then identified based on literature data and spectroscopic analysis. It was concluded that the two compounds were β-sitosterol and 2-(5′,6′-dimethoxy-3′,4′-methylenedioxyphenyl)-6-(3″,4″,5-trimethoxyphenyl)-dioxabiclo [3,3,0] octane. In addition, the two compounds were also reported to have antitumor activity with an IC_50_ value of 2.04; 1.34, 2.08, and 27.40 g/mL in the fractions of n-hexane, ethyl acetate, buthanolic, and methanolic extract, respectively [118] (Figure 6).

Arbain et al., 2018 isolated a 1.10 kg sample of *P. crocatum* Ruiz and Pav by using the maceration extraction method twice with methanolic solvent (5 L) for 48 h. Two new bicyclo [3.2.1] octanoid neolignans of the guianine type, crocatin A and crocatin B, together with the known compounds pachypodol and 1-triacontanol isolated from Indonesian *P. crocatum* Ruiz and Pav leaf. Its structure and configuration were determined by 1D- and 2D-NMR, MS spectroscopy, and single-crystal X-ray diffraction analysis [119] (Figure 7).

In a study conducted by Chai et al. (2021), 2.60 kg of dried leaves of *P. crocatum* Ruiz and Pav were isolated which were then extracted using the reflux method using methanol (5 L × 3 times) as a solvent. The isolation results reported that four bicyclo [3.2.1] octanoid neolignans were isolated from the methanolic extract of *P. crocatum* Ruiz and Pav. Neolignans were identified as pipcroside A, pipcroside B, pipcroside C, and crocatin B. In addition, this study by Chai et al., 2021 also provides the basis for further exploration of *P. crocatum* Ruiz and Pav and bicyclo [3.2.1] octanoid neolignans from the *Piper* plant as a new source of natural antineoplastic agents [120] (Figure 8).

### 4.2. Bioactivity of Piper crocatum Ruiz and Pav

The *Piperaceae* family is one type of plant that is often found in the surrounding environment and several types of plants in that family are classified as dicotyledonous plants. One of them that is often used by the community as a traditional medicinal plant is the *Piper* genus. It has more than 700 species spread throughout the world and commercial, economic, and medicinal importance. Many plant species of this genus have high potential for local and industrial uses, as well as applications in botanical pharmacy, pharmacognosy, and traditional medicine. The efficacy of the drug basically comes from several secondary metabolite compounds contained in the plant.

Secondary metabolites of the *Piper* genus, in addition to their unique structure, are also reported to have potential as bioactive compounds. Tests for the bioactivity of this genus have been carried out on both extracts and pure compounds. The isolation results support its use in traditional medicine (Table 1).

Like plants from other *Piper* genera, *P. crocatum* Ruiz and Pav also has some bioactivity, both from the level of extract, fraction and isolation results, and several instances of bioactivity of red betel have been reported. In the table below are some studies of isolation of *P. crocatum* Ruiz and Pav with various kinds of bioactivity of each (Table 2).

### 4.3. Antibacterial Activity of Red Betel Extract

One of the examples of bioactivity of *P. crocatum* Ruiz and Pav, which is the topic of this review, is antibacterial activity. Especially, the antibacterial activity of red betel against the bacteria *S. mutans*, *S. sangguinis*, *V. parvula,* and other bacteria found in the oral cavity that cause dental and oral health problems, one of which is dental caries. Therefore, the potential of red betel as an antibacterial agent can be understood by looking at several studies that have been reported. The table below shows data from previous research reports that reported the antibacterial ability of red betel leaf extract (Table 3).

In research conducted by Rizkita et al. (2017), the research procedure includes four stages, namely plant determination, betel leaf oil refining, identification of betel oil components, and betel oil activity test, then the two oils are compared. Further component identification was carried out by mass spectrometry. The results of mass spectrometry will obtain the mass spectrum of each peak detected on the GC chromatogram. The mass spectra analysis was based on the value of Similarity Index (SI), base peak, and the fractional trend of the mass spectra compared to the library mass spectra, namely WILEY229.LIB. It was reported that the isolation results from *P. betle* L. and *P. crocatum* Ruiz and Pav contain essential oils which consist of five main active compounds that have antibacterial properties. The test was carried out by applying the disc method. The media used was Mueller Hinton Agar media because in this medium *S. mutants* bacteria lived optimally. The agar media that had been planted with the test bacteria were filled with samples of green betel oil and red betel oil with concentration variations (100, 75, 50, and 25%), propylene glycol solvent as a negative control, and amoxicillin as a positive control (Figure 9) [13].

These compounds are terpenoid group compounds including camphene, sabinene, cariophilene, humulena, and germakron in green betel while the terpenoid compounds in red betel leaf include sabinene and mirsen. The antibacterial activity test of these compounds proved that there was an inhibition of the growth of *S. mutans* bacteria. Antibacterial compounds are thought to be able to inhibit the growth of Gram-positive bacteria by penetrating the cell wall, the cell wall of Gram-positive bacteria has a simple composition consisting of 60–100% peptidoglycan, which is made of *N*-acetyl glucosamine and *N*-acetyl muramate. The simple arrangement of the cell wall and the absence of an outer membrane causes antibacterial compounds to penetrate the cell wall and interfere with the cell wall biosynthesis process.

Sesquiterpene compounds have hydrophobic properties that cause disruption of the integrity of bacterial cells by reducing intracellular ATP reserves, lowering cell pH, being absorbed and penetrated into bacterial cells, then bacteria will experience precipitation and protein denaturation, and will lyse bacterial cell membranes. The difference in the concentration of the content contained in green betel leaf and red betel leaf contains 1.00–4.20% (*w*/*v*) essential oil yield, chavicol 7.20–16.70%, cavibetol 2.70–6.70%, and eugenol 26.80–42.50%. Meanwhile, the yield of red betel leaf was 0.73 (*w*/*v*), chavicol 5.10–8.20%, and eugenol 26.10–42.50%.

## 5. Conclusions

Medicinal plants of *P. crocatum* Ruiz and Pav have a significant role in applications of ethno-medicine. They contain secondary metabolites that have several examples of bioactivity, such as antioxidant, antimicrobial, antibacterial, antifungal, anti-inflammatory, and others. The bioactivity is influenced by the structure and functional groups of each secondary metabolite compound contained therein. Based on several research reports, it can be seen that *P. crocatum* Ruiz and Pav has considerable potential as an antibacterial agent in the treatment of oral health problems such as dental caries with several different methods. Secondary metabolites contained in *P. crocatum* Ruiz and Pav have their own mechanism to inhibit bacteria. This scientific finding is useful information for further drug research and development to find new potential antimicrobial agents.

## Figures and Tables

**Figure 1 molecules-27-02861-f001:**
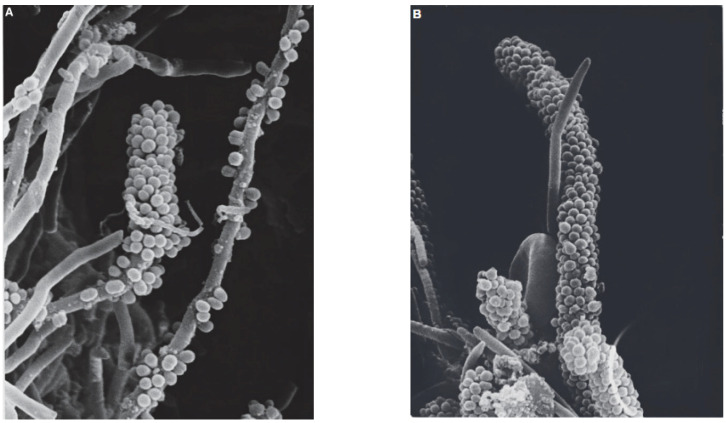
(**A**) Co-aggregation between *S. mutans* and filaments in developing dental biofilm; (**B**) typical corncob formation [30].

**Figure 2 molecules-27-02861-f002:**
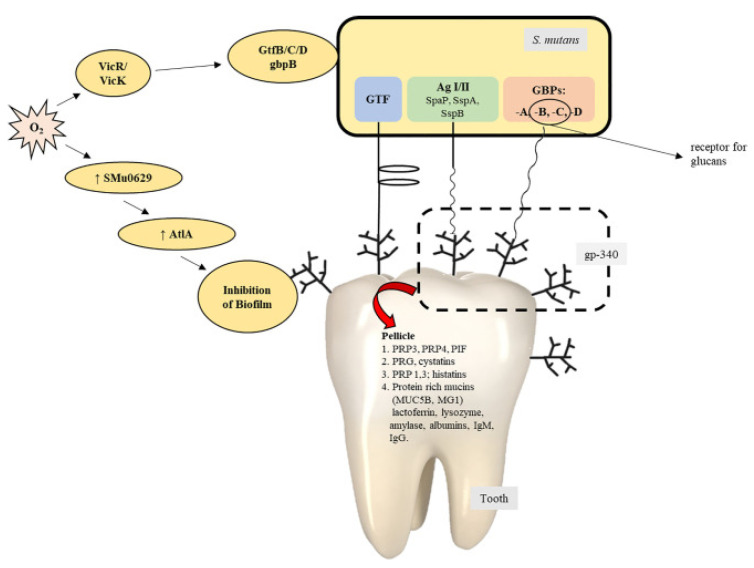
Contribution of *S. mutans* in the process of biofilm formation [39].

**Figure 3 molecules-27-02861-f003:**
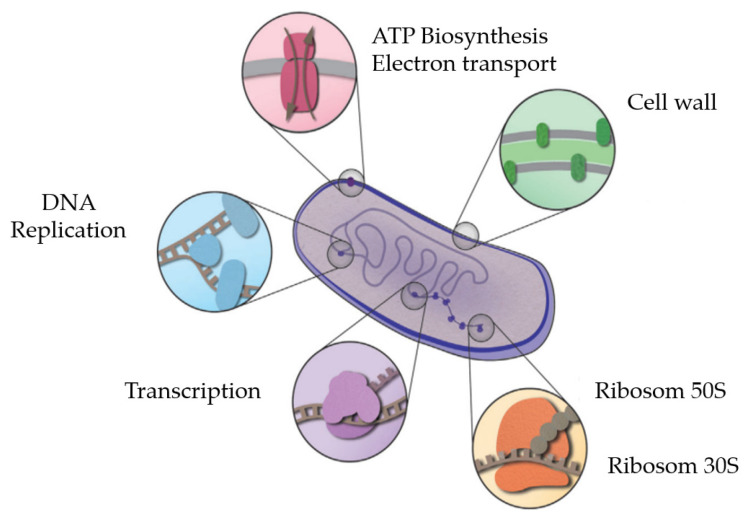
Pathway of inhibition of bacteria by antibacterial agents [73].

**Figure 4 molecules-27-02861-f004:**
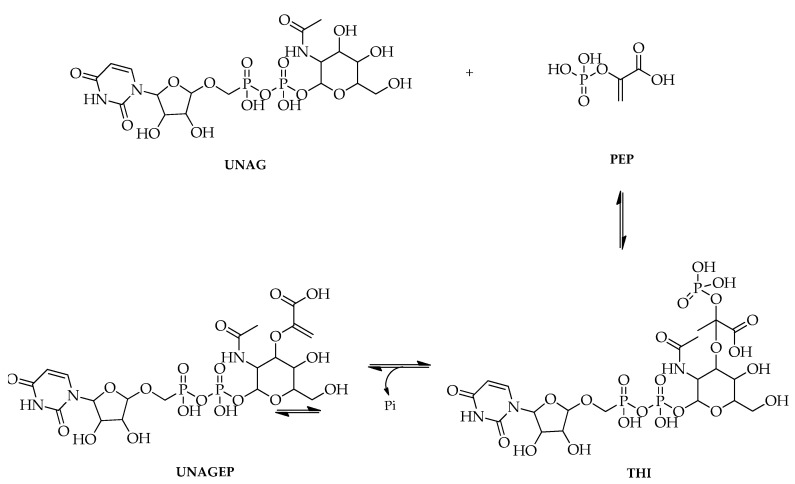
Catalytic reaction on the MurA enzyme [106].

**Figure 5 molecules-27-02861-f005:**
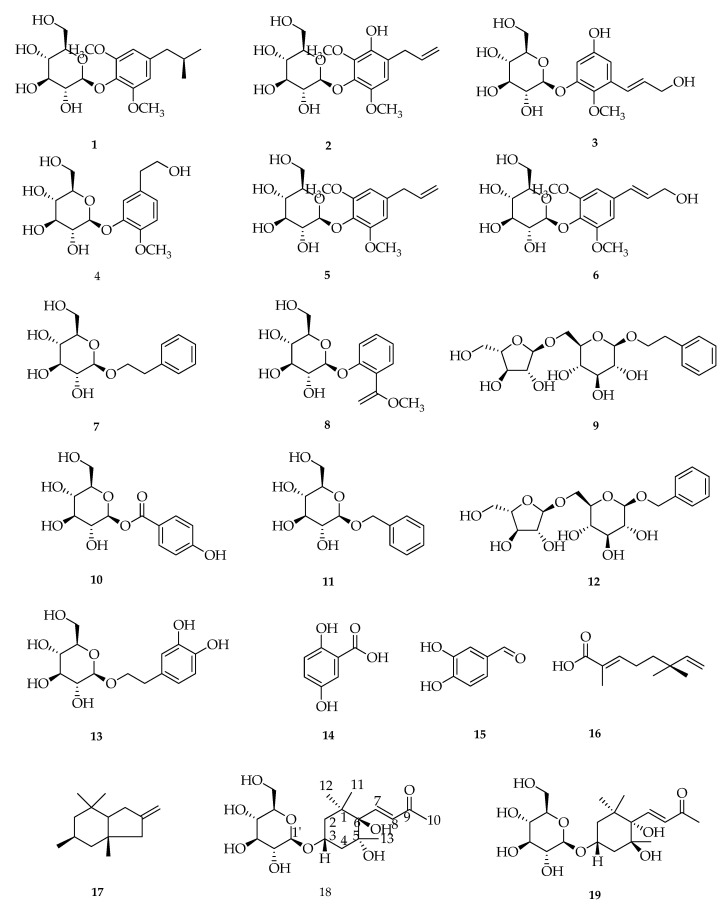
Compounds obtained from the methanol extract of red betel leaf. (**1**) (8*R*)-8-(4-hydroxy-3,5-dimethoxy)-propane-8-ol-4-*O*-β-D-glucopyranoside; (**2**) 4-Allyl-2,6-dimethoxy-3-hydroxy-1-D-glucopyranoside; (**3**) 3-[(1*E*)-3-hydroxy-1-propen-1-yl]-2,5-dimethoxyphenyl-D-glucopyranoside; (**4**) Cimidahurinin; (**5**) Erigeside II; (**6**) Syringe; (**7**) β-phenylethyl-β-D-glucoside; (**8**) Methylsalicylate-2-*O*-β-D-glucopyranoside; (**9**) Icariside D1; (**10**) 4-Hydroxybenzoic acid-D-glucosylester; (**11**) Benzyl-β-D-glucoside; (**12**) Phenylmethyl-6-*O*-α-L-arabinofuranosyl-β-D-glucopyranoside; (**13**) Hydroxytyrosol-1glucopyranoside (**14**) Gentisic acid; (**15**) Catechaldehyde; (**16**) (*S*)-Menthiafolic acid; (**17**) Ioliolide; (**18**) 5β,6β-dihydroxy-3α-(β-D-glucopyranosyloxy)-7*E*-Megastigmen-9-one; (**19**) (3*E*)-4-[(1*S*,2*S*,4*S*)-4-(β-D-glucopyranosyloxy)-1,2-dihydroxy-2,6,6-tri-methylcyclohexyl]3-buten-2-one; (**20**) (6*S*,9*S*)-roseoside; (**21**) Cuneataside E (**22**) *N*-trans-feruloyltyramine-4′-*O*-β-D-glucopyranoside; (**23**) Syringaresinol-β-D-glucoside; and (**24**) Vitexin 2″-*O*-rhamnoside.

**Figure 6 molecules-27-02861-f006:**
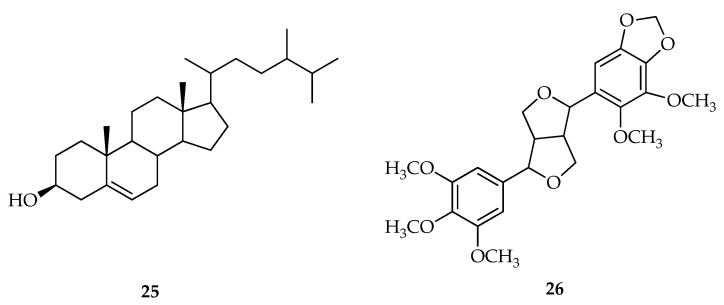
Compounds obtained from the methanolic extract of red betel leaf (*P. crocatum* Ruiz and Pav). (**25**) β-sitosterol and (**26**) 2-(5′,6′-dimethoxy-3′,4′-methylenedioxyphenyl)-6-(3″,4″,5-trimethoxyphenyl)-dioxabiclo [3,3,0] octane.

**Figure 7 molecules-27-02861-f007:**
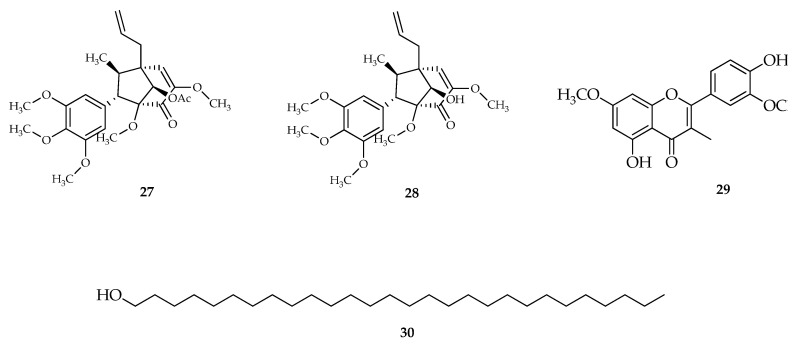
Compounds obtained from the methanolic extract of red betel leaf (*P. crocatum* Ruiz and Pav). (**27**) Crocatin A; (**28**) Crocatin B; (**29**) Pachypodol [4′,5-dihydroxy-3,3′,7-trimethoxyflavone]; and (**30**) 1-Triacontanol.

**Figure 8 molecules-27-02861-f008:**
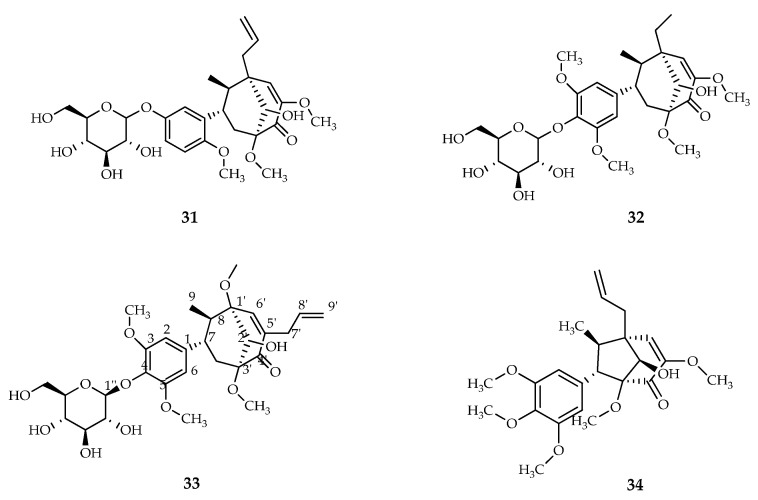
Compounds obtained from the methanolic extract of red betel leaf. (**31**) Pipcroside A; (**32**) Pipcroside B; (**33**) Pipcroside C; and (**34**) Bicyclo [3.2.1] octanoid neolignans.

**Figure 9 molecules-27-02861-f009:**
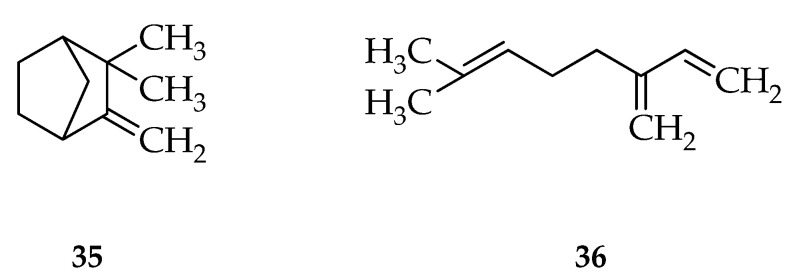
Structure of compounds of isolated red betel leaf oil. (**35**) Camphene and (**36**) Myrcene [13].

**Table 1 molecules-27-02861-t001:** Bioactivity of isolated *Piper* genus.

No.	Species	Secondary Metabolites	Plant Parts	Bioactivity	References
1	*P. betle*	Phenylpropanoid	Leaf	Antioxidant	Atiya et al., 2018 [121]
2	*P. terminaliflorum tseng*	Furfuran Lignan	All parts of plant	Anticancer	T. Liu et al., 2018 [122]
3	*P. chimonantifolium*	Flavonoids Steroids	Leaf	Antifungal	Lago et al., 2012 [123]
4	*P. montealegreanum*	Monoterpens Seskuiterpens	Twig		Da S. Alves et al., 2011 [124]
5	*P. hispidum*	Chalcones, Flavanone	Leaf	Antileishmanial	Ruiz et al., 2011 [125]
6	*P. maingayi*	Amida	Twig	Antibacterial	Hashim et al., 2019 [126]
7	*P. officinarum*	Phenylpropanoid Alkaloids Triterpene	Twig	Antioxidant	Salleh et al., 2014 [127]
8	*P. taiwanense*	Amida	Aerial	Antioxidant	Chen et al., 2017 [128]
9	*P. sarmentosum*	Flavonoids	Leaf	Antioxidant	Ugusman et al., 2011 [129]
10	*P. solmsianum C.*	Flavonoids	Twig	Antifungal	De Campos et al., 2005 [130]
11	*P. betle* L.	Terpenoid	Leaf	Antibacterial	Batubara et al., 2011 [131]
12	*P. betle* L.	Phenolic	Leaf	Antibacterial	Kurnia et al., 2020 [132]
13	*P. ningrum*	Alkaloid-piperidine	Fruit	Anticancer	Reshmi et al., 2010 [133]

**Table 2 molecules-27-02861-t002:** Bioactivity of isolated *P. crocatum* Ruiz and Pav leaves.

No.	Secondary Metabolites	Plant Parts	Bioactivity	References
1	Flavonoids Terpenoids Steroids	Leaf	Antitumor	Emrizal et al., 2014 [118]
2	2 flavonoids 2 monoterpenes 3 seskuiterpenes 17 Glucoside	Leaf	Anti-inflammatory	Li et al., 2019 [117]
3	12 Phenolic	Leaf	Hypoallergenic	Li et al., 2019 [134]
4	Bicyclo[3,2,1]Octanoid Neolignane	Leaf	Pyruvate dehydrogenase inhibitors	Chai et al., 2021 [120]
5	Essential Oil	Leaf	Antibacterial	Rizkita et al., 2017 [13]

**Table 3 molecules-27-02861-t003:** Antibacterial activity methods of red betel extract (*P. crocatum* Ruiz and Pav).

No.	Compounds	Types of Bacteria	Methods	References
1	Flavonol Chalcone Anthocyanins	*S. mutans*	The Kirby–Bauer method of the disc diffusion test combined with UV irradiating treatment was used. The results showed the diameter of the inhibition zone (15.00 ± 0.05) mm for 10 watt and (15.96 ± 0.05) mm for 15 watt.	Dyah Astuti et al., 2020 [135]
2	Alkaloids Steroids Tannins	*B. subtilis* *P. aeuruginosa*	Antibacterial activity was tested using the well method. Inhibited the growth of *B. substilis* and *P. aeruginosa* bacteria but the activity was weak, the inhibition zone was < 5 mm.	Puspita et al., 2019 [136]
3	Flavonoid Saponin Tannins Phenolic	*Staphylococcus epidermidis*	Bacterial test was carried out using the well method, extract concentrations of 50 and 100% could inhibit the growth of *S. epidermidis.*	Januarti et al., 2019 [137]
4	Tannins	*Staphylococcus aureus*	Tests using the well method can inhibit *S. aureus* bacteria. Maceration extraction technique to get the average inhibition zone of 12.30 mm.	Soleha, 2018 [138]
5	Flavonoids Alkaloids Tannins Essential oil	*Porphyromonas gingivalis* *S. viridians*	The antibacterial test was carried out using the well method, the inhibition zone on *P. gingivalis* was 10.34 mm while *S. viridians* was 8.42 mm.	Pujiastuti et al., 2015 [139]

## Data Availability

The study did not report any data.

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
