# Peer review of "Phytochemical Profile of Antibacterial Agents from Red Betel Leaf (Piper crocatum Ruiz and Pav) against Bacteria in Dental Caries"

_molecules, 2022, doi:10.3390/molecules27092861_

Round 1

Reviewer 1 Report

Although the content of the review is interesting, the English has to be improved. There were some obvious grammatical errors (listed below). More detail could also be given regarding those compounds that are active against oral bacteria, even though the authors focus on Streptococcus mutans. There is not much critique of the known literature in the manuscript, e.g. can the authors compare the efficacies of the various compounds from Piper crocatum Ruiz & Pav (Table 3)?

Comments:

  1. In the abstract, Streptococcus and Piper should be defined on first use. This is because Staphylococcus can also be abbreviated to S., as can Pseudomonas and Porphyromonas to P..
  2. Lines 58 & 60 - Candida, not Caaandida; nucleatum, not Nucleatum.
  3. Lines 80-81 - glycolysis breaks down glucose, not polymerization.
  4. Line 87 - lime phosphate?  Do the authors mean hydroxyapatite?
  5. Line 112 - hydroxyapatite is the major mineral in enamel.
  6. Line 160 - bactericidal means it will kill bacteria, not can.
  7. Line 167 - N-acetylmuramic acid, not MurA. MurA is the enzyme.
  8. Figure 3 - The readers might not be able to understand Bahasa Indonesia - maybe the authors can translate to English?
  9. Lines 227-230 - NH, F and FR are only used once, so why have the abbreviations?
  10. AlK(SO4)2 is actually alum (or potassium alum), not Aluminum (the metal).
  11. The name of the plant is red betel, not Red betel or red Betel. Perhaps the authors can be more consistent?
  12. Line 273 - MeOH is methanol.
  13. Line 299 - Emrizal et al. is [121], not simplicia.
  14. Line 332 - Chai et al., not Chi.
  15. Lines 342-349 - maybe the authors would like to combine the sentences into a longer sentence, rather than having so many "of the genus Piper" in several short sentences?
  16. Table 3 - what does the watts refer to? This is a study combining UV irradiation in combination with the extracts. The authors do not provide these important details, especially in the table directly related to the aim of the review.
  17. Lines 390-393 - this sentence does not make sense. Can you make agar filled with 100% betel oil? (maybe a photo of the inhibitory activity might be helpful to the reader.)

Author Response

Note: the words or sentence fixed are marked with yellow-coloured highlight.

Reviewer 1

Evaluation for authors

  1. In the abstract

Answer:

Thank you for your correction and suggestion.

The name “Streptococcus and Piper” have been added on the first use.

  1. Line 58 & 60

Answer:

Thank you for your correction.

The authors made a mistake in typing and incorrect words has been changed.

  1. Line 80-81

Answer:

Thank you for your correction.

The term “polymerization” has been removed and changed with glycolysis.

The authors also add the definition of glycolysis to give understanding easily to the reader.

  1. Lines 87

Answer:

Thank you for your correction.

Yes, we do.

The lime phosphate (phosphate of lime) means hydroxyapatite, and it has been changed with hydroxyapatite to be more common.

  1. Lines 112

Answer:

Thank you for your correction.

It has been changed with “the major”.

  1. Lines 160

Answer:

Thank you for your correction.

It has been changed with “will kill”.

  1. Line 167

Answer:

Thank you for your correction.

The incorrect words has been changed with the correct words suggested become N-acetylmuramic acid.

  1. Figure 3

Answer:

Thank you for your correction and suggestion.

The author have changed the caption on the figure in English.

  1. Lines 227-230

Answer:

Thank you for your correction and suggestion.

Those have been changed with nicomethanol hydrofluoride, fluoride, fluoride-resistant and not be abbreviated.

  1. AlK(SO4)2

Answer:

Thank you for your correction.

Aluminum has been removed and changed to AlK(SO4)2 only.

  1. red betel

Answer: Thank you for your correction.

The use of the name of plant has been homogeneous become “red betel” throughout the manuscript, except on the first sentence and the title.

  1. Line 273

Answer:

Thank you for your correction.

It has been changed become methanol and not be abbreviated throughout the manuscript.

  1. Line 299

Answer:

Thank you for your correction.

It has been removed.

  1. Line 332

Answer:

Thank you for your correction.

It has been fixed in the manuscript.

  1. Lines 342-349

Answer:

Thank you for your suggestion.

The authors have combined into a longer sentence, so that decreasing the phrase “the genus Piper”.

  1. Table 3

Answer:

Thank you for your correction.

Watt in the table refers to the unit of the power electricity of UV irradiation as a light source used on that study.

The authors do not provide the detail information because just want to show the antibacterial activity method of P. crocatum Ruiz & Pav used on that study.

Table 3 just shows antibacterial activity of secondary metabolites of P. crocatum Ruiz & Pav using different methods against several bacteria.

  1. Lines 390-393

Answer:

Thank you for your correction.

100% means concentration variation of sample (red betel oil) and does mean how much sample (red betel oil) that will be filled into agar.

Rizkita et al. [13] does not provide the photos.

Reviewer 2 Report

The Review article entitled; Phytochemical Profile of Antibacterials Agents from Red Betel 2 Leaf (Piper crocatum Ruiz & Pav) against Streptococcus mutans 3 Bacteria in Dental Caries is good work suitable for publication in molecules after minor revision as the review loss the conclusion part.

The authors should add the conclusion part.

Author Response

Note: the words or sentence fixed are marked with yellow-coloured highlight.

Reviewer 2

Evaluation for authors

The Review article entitled; Phytochemical Profile of Antibacterials Agents from Red Betel 2 Leaf (Piper crocatum Ruiz & Pav) against Streptococcus mutans 3 Bacteria in Dental Caries is good work suitable for publication in molecules after minor revision as the review loss the conclusion part. The authors should add the conclusion part.

Answer:

Thank you for your great corrections, suggestions and responses.

The authors have added the conclusion section.

Reviewer 3 Report

The work needs a series of corrections before being accepted by the journal.

"Phytochemical Profile of Antibacterial Agents from Red Betel Leaf (Piper crocatum Ruiz & Pav) against Streptococcus mutans Bacteria in Dental Caries." The title should be changed to a more general one because it is expected that only Stretococcus mutant bacteria will be discussed and the antibacterial activity of Red betel leaf on many more bacteria will not be discussed. Phytochemical Profile of Antibacterial Agents from Red Betel Leaf (Piper crocatum Ruiz & Pav) against Bacteria in Dental Caries.

Line 50: Streptococcus mutans is a Gram-positive bacterium. Change to S. mutans mutans…
Line 75: of S. mutans begins. Correct italics “of”

3.2 Antibacterial Mechanism of Secondary Metabolic Compounds
3.2.1 Phenol
The mechanism of phenol as an antibacterial agent acts as a toxin in the protoplasm, damaging and penetrating the wall, causing the function of selective permeability, active transport, and protein composition control, so that bacterial cells become deformed and lysed [76-78] . This section is not introduced in an adequate way, it seems that it is disconnected from the previous topic. Some introductory paragraphs should be included that talk about the presence of phenolic compounds in general in plants (total phenolic content, total falvanoids content….) and their biological properties, focusing on antibacterial activity. There we could already talk about 3.2.1 Phenol compounds… but it should be more extensive.

Lines 270-272: 4.1 Isolation of Secondary Metabolites of Piper crocatum Ruiz & Pav Several studies reported the isolation of P. crocatum Ruiz & Pav by many methods. Li et al., 2019 isolated 2.6 kg of dried red betel leaf samples, then extracted by reflux method using MeOH solvent (5L×3times). The results of the isolation of P. crocatum leaves.. To establish a homogeneous abbreviation criterion for the entire manuscript for "Piper crocatum Ruiz & Pav". 1) P. crocatum Ruiz & Pav, or 2) P. crocatum.

Lines 299-308: Another isolation method was carried out by Emrizal et al., 2014 simplicia P. crocatum Ruiz and Pav as much as 0.841 kg were extracted at room temperature with MeOH solvent to obtain a crude methanol extract of 253.27 g (30.1153% ) after which the extract was evaporated and proceed to separate the components of the compound. The results of the isolation obtained two compounds from the P. crocatum plant which were then identified based on literature data and spectroscopic analysis, it was concluded that the two compounds were β-sitosterol and 2-(5',6'-dimethoxy-3 ',4'-methylenedioxyphenyl)-6- (3”,4”,5-trimethoxyphenyl)-dioxabiclo [3,3,0] octane. In addition, the two compounds were also reported to have antitumor activity with an IC50 value of 2.04; 1.34, 2.08 and 27.40 307 g/mL in the fractions of n-hexane, ethyl acetate, butanol, and methanol extract, respectively [121] (Figure 6). A single and homogeneous criterion must be established throughout the manuscript to express the numerical figures regarding the use of decimals. There are figures with one decimal, with two, with three and up to four decimals. Please correct the entire manuscript including tables.

Best regards, 

Author Response

Note: the words or sentence fixed are marked with yellow-coloured highlight.

Reviewer 3

Evaluation for authors

  1. Title

Answer:

Thank you for your corrections and suggestion.

The authors have changed the title with the title suggested.

  1. Line 50

Answer:

Thank you for your suggestion.

It has been changed to S. mutants.

  1. Line 75

Answer:

Thank you for your correction.

It has been fixed.

  1. Section 3.2

Answer:

Thank you for your correction and suggestion.

The authors have added several sentences on the previous to connect or introduce the title of topic with the content (3.2.1. etc).

  1. Lines 270-272

Answer:

Thank you for your correction and suggestion.

It has been fixed become a homogeneous abbreviation criterion (P. crocatum Ruiz & Pav) throughout the manuscript, except on the first use and the title, it still uses Piper crocatum Ruiz & Pav.

  1. Lines 299-308

Answer:

Thank you for your suggestion.

It has been fixed become a homogeneous criterion throughout the manuscript using two numbers on the decimal.

Round 2

Reviewer 1 Report

Species names have to be italicised.

Reviewer 3 Report

Thanks to the authors for accepting the suggestions. Please correct the names of the microorganisms. They should all be in italics. I have no more comments.
